# Apoptosis in Type 2 Diabetes: Can It Be Prevented? Hippo Pathway Prospects

**DOI:** 10.3390/ijms23020636

**Published:** 2022-01-07

**Authors:** Agnieszka Kilanowska, Agnieszka Ziółkowska

**Affiliations:** Department of Anatomy and Histology, Collegium Medicum, University of Zielona Gora, Zyty 28, 65-001 Zielona Gora, Poland; a.ziolkowska@cm.uz.zgora.pl

**Keywords:** type 2 diabates, Hippo pathway, MST, LATS, YAP, apoptosis, caspase, β cells, pancreatic islets

## Abstract

Diabetes mellitus is a heterogeneous disease of complex etiology and pathogenesis. Hyperglycemia leads to many serious complications, but also directly initiates the process of β cell apoptosis. A potential strategy for the preservation of pancreatic β cells in diabetes may be to inhibit the implementation of pro-apoptotic pathways or to enhance the action of pancreatic protective factors. The Hippo signaling pathway is proposed and selected as a target to manipulate the activity of its core proteins in therapy-basic research. MST1 and LATS2, as major upstream signaling kinases of the Hippo pathway, are considered as target candidates for pharmacologically induced tissue regeneration and inhibition of apoptosis. Manipulating the activity of components of the Hippo pathway offers a wide range of possibilities, and thus is a potential tool in the treatment of diabetes and the regeneration of β cells. Therefore, it is important to fully understand the processes involved in apoptosis in diabetic states and completely characterize the role of this pathway in diabetes. Therapy consisting of slowing down or stopping the mechanisms of apoptosis may be an important direction of diabetes treatment in the future.

## 1. Introduction

Diabetes mellitus is a heterogeneous disorder with multiple etiologies [1,2]. The most prevalent in adults is type 2, a chronic metabolic illness with increasing medical and financial costs [3,4]. The incidence of diabetes in 2015 was 8.8%, and in 2040 it will probably increase to 10.4% [5]. Its most characteristic symptom is a high level of glucose as a result of insufficient insulin for the organism needs [6]. On the other hand, symptoms are consequences of genetic and/or lifestyle factors and are largely related to obesity [7,8]. β cell dysfunction and worsening insulin resistance over time lead to deterioration of glycemic control and, as a result, need for more intensive pharmacotherapy [9,10]. Over time, this is usually manifested by a complete inability of the endogenous islet resulting from a loss of β cells [10,11,12], which also includes impaired proliferation and dedifferentiation [13,14]. The metabolic stress observed in type 2 diabetes can induce the pro-inflammatory cytokines, tumor necrosis factor-α (TNF-α) and interleukin-6 (IL-6), activating a series of complex processes leading to apoptotic death of pancreatic β cells [15,16]. Glucotoxicity, lipotoxicity [17,18,19,20], endoplasmatic reticulum (ER) stress [21,22], oxidative stress [23,24], islet amyloid polypeptide (IAPP) [25], and inflammation [15,26] and are responsible for the loss of pancreatic mass, leading to progressive β-cell failure of the pancreas in type 2 diabetes [19,27,28]. The destabilization of the endoplasmic reticulum [29] poses a huge threat to its function in the context of the secretory pathway. Saturated fatty acids (e.g., palmitate) have been shown to contribute to apoptosis through the induction of reactive oxygen species (ROS) and the ER stress pathway [30]. A similar effect was also observed in chronic hyperglycemia [31]. Induced ER stress is manifested by an imbalance between the ability to fold proteins in relation to the needs of the cell, leading to disturbances in homeostasis [32]. Functional disproportion causes the deposition of unfolded or misfolded proteins in the lumen of the ER [33]. Unfolded protein response (UPR) is an adaptive mechanism to restore homesotase [31]. It involves, inter alia, increasing the functional capacity of the organelle through the transcriptional regulation of folding and chaperone enzymes, as well as increasing the size of the ER. Usually, a reduced influx of newly synthesized proteins is observed as a result of the inhibition of protein translation and mRNA degradation. The ER stress transducers present in the membrane oversee this process through two luminal domains. One of them is involved in the recognition of unfolded proteins. The second cytoplasmic domain sends signals to the cytosol and nucleus if the stabilization measures taken are ineffective [34]; the same communication system is used to initiate the apoptotic pathway [33]. The apoptotic pathway involves two protein families, including the Bcl-2 protein family (an acronym for B-cell lymphoma 2 gene) and caspases [35] which is the basis for distinguishing two pathways of apoptosis [36]. The first is called the “external” pathway (related to Fas or TNF death receptors (TNFR)) [25]. The second is defined as the internal pathway (associated with the mitochondria) and carried out by the involved proteins from the Bcl-2 family [29,37]. The first of these is activated after the ligation of death receptors (e.g., CD9, TNFR) on the cell surface. The expression of Fas in β cells (external apoptotic pathway), is induced by cytokines and tumor necrosis factor alpha (TNF-α), initiating the activation of the transcription factor, nuclear factor κb, and signal transducer and activator of transcription 1 (STAT-1) [38], leading to the activation of a caspases cascade, which implements the effector mechanism. The second of the distinguished mechanisms emphasizes the role of mitochondrial processes. A characteristic feature of the mitochondrial apoptosis mechanism is the release of cytochrome c into the cytoplasm from the mitochondria. The effect is the activation of initiator caspase-9, resulting in the cleavage of executive caspases-3/7 [39]. Proteins from the Bcl-2 family are involved in the integrity of mitochondria and are a complex group of proteins [40,41].

Changes in the treatment of type 2 diabetes should be based on the effective inhibition of the loss of insulin-secreting cells, while restoring their full functionality. Current treatments for type 2 diabetes only alleviate symptoms without ruling out the underlying causes. The above assumptions require a complete analysis of the signaling pathways related to β cell loss. The intense course of apoptosis in type 2 diabetes ultimately results not only in the loss of islet mass, but is also associated with morphological changes, including cell rounding, vesicle formation, and chromatin condensation [14,42]. Although the mechanism of apoptosis has been identified at many levels (expression of apoptosis genes, triggering, initiation of signaling cascades) [43], in the treatment of type 2 diabetes, targeted action on its arrest has not been introduced. Intensive research in recent years shows a promising prognosis and probably the therapeutic basis for the use of the Hippo pathway [44,45,46]. It was proven that the general components and functions identified in the Hippo pathway are involved in regulating organ size by maintaining a balance between apoptosis, proliferation, and cell/tissue regeneration [47,48]. The final number of cells in an organ is the result of a balanced act of cell proliferation and cell death (apoptosis) [49] and is dependent on the course of the Hippo pathway [50]. Primary studies have shown that the abnormal expression of the Hippo signaling pathway in tested peripheral tissues may initiate type 2 diabetes and be associated with the gut microbiota, as well as with the aging process [51]. The identification of those signaling pathways on which the functioning and survival of pancreatic β cells depends is a prerequisite for a full understanding of the mechanisms related to dysfunction and will be the basis for introducing new solutions in therapy [43]. The aim of this review is to delineate the role of the Hippo signaling pathway in pancreatic β cell apoptosis in diabetes, and the possibilities it presents in eliminating the consequences of diabetes in both cellular and systemic terms.

## 2. The Hippo Pathway—How Exactly Does It Work?

The Hippo pathway is a complex communication network whose signals are modulated by numerous kinases/phosphatases and adapter proteins. Due to the versatility and sensitivity of the mechanism, its core cross-reacts with other pathways. Thus, they ensure appropriate responses to exogenous and endogenous stimuli [52]. It is also defined as an important regulator of organ size control and embryo development through the regulation of cells: proliferation, apoptosis, and differentiation. Their sum creates the conditions for maintaining tissue homeostasis [53]. It was first disclosed in *Drosophila melanogaster* and named after Hippo kinase [54]. The components of this pathway are evolutionarily conserved [55] and, importantly, preserved in all mammals [56]. The essential elements of the Hippo pathway, referred to as the core of the Hippo pathway, are the mammalian sterile 20-like protein kinase 1 and 2 (MST1/2), kinase cascade, and large tumor suppressors 1 and 2 (LATS1/2) [45]. The pathway is co-created by the MST1/2 kinases, two Salvador adapter proteins (Sav1 scaffold protein) and MOB1A or MOB1B, Warts (Wts/large tumor suppressor 1 (LATS1) and mammalian LATS2), two Yorkie homologs of the Yes protein (YAP in mammals), a transcriptional coactivator from the PDZ binding motif (TAZ), and the transcription factor TEA 1 (TEAD1) to TEAD4 (Figure 1) [57]. The MST-LATS-YAP1/TAZ-TEAD interaction axis is the core of the mammalian Hippo pathway and is referred to as the canonical Hippo pathway [43,55].

The course of this pathway depends on the activity of its own elements, creating the conditions for observing its functioning in two variants: when the Hippo signaling pathway is on or off. Further, the course is related to the activation of MST1/2, which, together with the adaptive Sav1 protein, is involved in the capture and phosphorylation of NDR LATS1/2 kinases. The activation of LATS1/2 by phosphorylation and the binding of MOB1A/B phosphorylates inactivate major effectors of the Hippo pathway, including the transcription coactivators associated with the Yes 1 protein (YAP), and the PDZ binding motif transcription co-activator (TAZ). The mechanism of action of MST1/2 and LATS1/2 is an example of the negative regulation of the activity of YAP and TAZ by promoting their retention and/or degradation in the cytoplasm [43]. Phosphorylated YAP and TAZ are recognized by 14-3-3 protein, which is sequestered (YAP/TAZ) to the cytoplasm or degraded in a ubiquitin-proteasome dependent manner [58], which results in silencing (induced by the Hippo pathway) of gene transcription [59]. The result is a reduction in the expression of YAP-dependent target genes, including anti-apoptotic and pro-survival genes [60] (Figure 1). It has been shown that genetic defects in the performance of the Hippo pathway in mice generate sustained tissue growth and can lead to cancer [61]. Conversely, when the Hippo pathway is off, its MST1/2 and LATS1/2 kinases are inactive. Unphosphorylated YAP and TAZ migrate to the nucleus. There, they initiate the transcription and expression of genes responsible for proliferation, differentiation, and apoptosis by interacting with TEAD1-4 and other transcription factors [54]. In studies where the Hippo pathway has been shown to be inactive (i.e., turned off), the transcriptional YAP and TAZ coactivators are hypophosphorylated and therefore translocate to the nucleus and induce their gene expression programs. It has been proven that many pathways converge in the YAP and TAZ registers, using their balancing in the nuclear/cytoplasmic region, which in turn translates into transcriptional activity [62]. They work by recruiting the transcription machinery for selected transcription factors, the best-known of which is the TEAD family [59]. It is therefore considered responsible for the general phenotypes controlled by Hippo signaling.

The overall Hippo signal path is complex and it is obvious that YAP1/TAZ, MST1/2, and LATS1/2 are regulated by other molecules and pathways that control various aspects of cell life [63].

## 3. The Mechanism of Apoptosis in Type 2 Diabetes and the Role of the Hippo Pathway

Many studies emphasize that pancreatic β cells are extremely sensitive to oxidative stress due to the low level of antioxidant enzymes (catalase, glutathione peroxidase and superoxide dismutase) [23]. The involvement of the Hippo pathway in the regulation of the apoptotic process resulting from the metabolic changes associated with type 2 diabetes has been documented on the basis of the contribution of its individual elements, including MST1 [64], Merlin (Nf2) [65], LATS2 [46,66], and YAP [65,67].

The regulation and physiological functions of the kinases that make up the core of the Hippo pathway in mammalian cells are not fully understood. It was confirmed that apoptosis activated by various extracellular stimuli promotes the apoptosis of cells by MST1/MST2, and caspase-catalyzed cleavage generates a highly active catalytic fragment, which is also associated with the process of programmed death [68]. The modulation of several signaling pathways and cellular organization is dependent on the Hippo core kinases, especially in the area of cell polarity and the actin cytoskeleton [69]. MST or its truncated form induce morphological changes in cells, some features of which correspond to the morphology of apoptotic cells in vitro: rounding, detachment from the substrate, or fragmentation and condensation of the nucleus [70]. Moreover, the overexpression of these kinases has been shown to induce apoptosis in a variety of cellular environments that are associated with the activation of stress-related pathways [45].

### 3.1. Upstream Regulators of Hippo Pathway 

#### 3.1.1. Merlin (Neurofibromin 2, Nf2)

The tumor suppressor gene, NF2, encodes the protein, 4.1, Ezrin, Radixin, and the Moesin (FERM) domain-containing protein Merlin (Nf2) [71]. Nf2 protein is highly conserved in mammals and regulates the Hippo signaling pathway through MST1/MST1 and LATS1/LATS2 in β cells (Figure 1) [72,73]. It acts as an initial activator of the Hippo pathway, thus exerting a real influence on the regulation of cell growth and proliferation [71]. Two mechanisms of the initiation of Hippo signaling by Merlin have been described. One is to initiate the pathway with MST-Sav activation followed by MST1/2 dependent LATS1/2 phosphorylation. The second starts with the recruitment of LATS1/2 to the membrane [74]. Merlin silencing, especially in diabedogenic conditions (glucotoxic, glycolipotoxic and pro-inflammatory), brings many benefits to β cells, protecting them against apoptosis. Moreover, both diabetogenic conditions and the introduced changes related to the loss of Merlin function in pancreatic β cells reduced only the phosphorylation of LATS1/2, but not of MST1/2 [65,73].

#### 3.1.2. Protein RASS

It has been shown that both MST1 and MST2 can form a constitutive homodimer via the SARAH domain. The autophosphorylation of the activation loop (T183 for MST1 and T180 for MST2) is the basis of kinase activation [52]. MST has been shown to interact with members of the Ras family of association domains of the MST/Hippo pathway (RASSF). The RASSF family of proteins consists of two subclasses, C-RASSF (RASSF1–6) and N-RASSF (RASSF7–10) and carry a common domain of the Ras-association. C-RASSFs, as regulatory proteins, are characterized by a C-terminal coiled-coil motif-SARAH domain (Salvador/RASSF/Hippo) [75], as is Sav1 [76]. Briefly, a heterodimer is formed from the MST1/2 domain of SARAH together with RASSF’s SARAH [77], while a heterotetramer is formed from Sav1 SARAH. The Sav1 junction and the RASSFs binding to MST1/2 are mutually exclusive. It is through this SARAH domain that a set of RASSF effectors form a complex with another MST1 protein kinase [76]. This aspect is important due to the creation of conditions for dimerization, the key mechanism of interaction between MST kinases and RASSF proteins, which is crucial currently. Dimerization is the basis for RASSF to participate in the regulation of the catalytic activity of MST kinases [78]. It has already been established that RASSF proteins feature several distinct domains and act as adapter proteins in many important biological processes, e.g., pro-apoptotic pathways, cell cycle, and cytoskeleton regulation [79]. Interaction between some members of RASSF and various Ras proteins has been noted [79]. Ras association is likely to locate RASSF and MST kinases in the plasma membrane, bringing the MST kinase domains in close proximity to trigger phosphorylation transactivation, driving the MST/Hippo pathway and cellular apoptosis [63,75,80]. The binding time between MST1/2 and RASSF allows the Hippo pathway to respond to and integrate a variety of cellular signals [76].

## 4. Core Proteins of the Hippo Pathway as a Therapeutic Target in Diabetes

### 4.1. The Serine/Threonine Kinase, Mammalian STE20-like Kinase 1/2 (MST1/2 Kinase) 

MST1 and MST2 are the canonical core of Hippo signaling in mammals [81]. Their action has been repeatedly confirmed and is largely related to the control of organ size [82,83,84]. In humans, they consist of 487 and 491 amino acids, respectively [85]. Moreover, they regulate not only cells’ architecture and polarity, but also proliferation, homeostasis, and death [52,81].

To date, five MST kinases have been described in mammals: MST1 (or STK4), MST2 (or STK3) Kinase II (GCK II) and MST3 (or STK24), MST4 (or STK26), and YSK1 (STK25/SOK1), representing the central reproductive kinase III (GCK III) [86]. Both MST1 and MST2 are closely related to GC class II kinases (Ser/Thr protein) [87]. An autoinhibition segment and a unique coiled-coil domain-SARAH [77] mediating the dimerization process are located between the N-terminal and the C-terminal regulatory domains [86,87,88]. SARAH domains (~50 amino acids in length) mediate the homo- and heterodimerization of MST1/MST2 because they contain RASSF effectors that bind to another protein kinase, MST1 [72]. Moreover, they are responsible for promoting protein-protein interactions with, among others, SARAH domain proteins [89]. It should be emphasized that MST1, as the core kinase of the Hippo pathway, is expressed in full length, e.g., in pancreatic β cells [43]. Scientific research contributed to the recognition of two caspase cleavage sites (D326 and D349) in the MST1/2 region between the kinase and the auto-inhibiting domain. Both internally arranged domains are removed by caspase cleavage, and their elimination activates the core protein during apoptosis [90,91]. Thus, MST1 takes two forms: a full-length 54 kDa protein, and a 36 kDa caspase cleaved version [92]. MST1 isoforms show different kinase activity (among themselves) with respect to substrates [93]. The caspase cleaved MST1 fragment is not only more active than the full-length because it lacks an autoinhibition domain, but also because it features different substrate selectivity. Truncated MST1 reduces the affinity for the forkhead box O of the transcription factor (FOXO), while increasing the affinity for the protein of the nucleosomal component, histone H2B [52,89]. The removal of the above domains increases its activity by about nine-fold compared to wild-type MST1 [88]. Moreover, it was found that it contributes to the formation of a positive feedback in maintaining the apoptotic signal through caspase activation, as well as the additional stimulation of MST1 cleavage [45]. The implementation of various apoptotic pathways, the course of which is associated with an increase in MST1 activity [43], has also been observed; therefore, this kinase is considered a critical regulator of apoptotic death and β cell dysfunction [45]. The adopted scheme of consequences related to the increased activity of MST1 in the diabetic environment (caused by cellular stress of various etiologies, including proinflammatory cytokines, glycolipotoxicity and oxidative stress) assumes the activation of mitochondria-dependent apoptotic processes, mainly by targeting the mitochondrial pro-apoptotic protein-Bcl-2 protein 11 (BIM), which causes changes in Bax/Bcl-2, release of cytochrome c, cleavage of caspase 9 and -3, and consequent cell death [45]. Both human and mouse pancreatic islets, as well as INS-1E cells exposed in vitro to diabetic conditions (complex states-more information [64]) have been found to correlate with increased MST activation (in all cases) and β-cell apoptosis [64]. Ardestani et al. [64] indicate that it is an important apoptotic molecule, induced in various diabetogenic states that are directly related to impaired survival and Langerhans islet cell dysfunction in diabetes. Based on this, it can be assumed that MST1 is most likely a common component of various signaling pathways leading to apoptosis. Pivotal studies showed that overexpression of MST1 in both human islets and INS-1E cells resulted in their apoptosis via caspase-3 cleavage and autophosphorylation (pMST1-T183). MST1-induced apoptosis occurs through a mitochondrial-dependent pathway. The expression profile of internal death pathway regulators containing only BH3 showed the highest induction of BIM. This points to a strong relationship (in human pancreatic islets) between MST1 and BIM, which is regulated by the c-jun N-terminal kinase (JNK) and AKT [64,94,95]. Under antagonistic conditions, silencing (80%) of MST1 in human pancreatic islets reduced the course of apoptosis, and a reduced BIM value was also reported in diabetic states [64]. Caspase-3 and JNK initiate a vicious circle of the pro-apoptotic signaling cascade in the β cell by interacting with MST1 as both upstream activators and downstream targets. The cleavage of MST1 is induced by various apoptotic signals, in vitro, including signals from the Fas receptor and TNFα [64,90]. It was confirmed that Fas, caspase and TNFα can induce apoptosis via MST1 kinase. MST1 and MST2 have been observed as direct substrates of caspase-3 in both in vivo and in vitro studies. They have been shown to act as up-and-down activators in the cell towards caspase activity, regulating its death receptor-induced activation. Both kinases are proteolytically activated by a specific cleavage and are involved in the Fas-mediated apoptosis process. The cleaved form of MST or its overexpression leads to more dramatic changes in β-cell morphology and is similar to those induced by PAK2 (serine-threonine protein kinase) [90,96]. In summary, under resting conditions, MST1 is localized exclusively in the cytoplasm. Caspases mediate the cleavage of MST1, resulting in the removal of the C-terminal inhibitory domain and the transfer of the active catalytic fragment to the nucleus [43,90,92] and the change in localization may be associated with a different offer of substrates for phosphorylation [91]. On the other hand, the ablation of MST1 revealed resistance to induced apoptosis by TNFα, Fas, and IFNγ [97,98,99]. In mice model of the deletion of MST1 (knockout mice MST1-/-) with diabetes induced experimentally (MLD-STZ) the loss of β cells was not observed. An analysis of the architecture of the pancreatic islets of these animals revealed preserved structure, density, size, and mass. It is also interesting that the β cells retained their function. In summary, genetically induced MST1 deficiency normalizes glycemia, restores the function of β cells, and prevents the development of diabetes [64].

MST1 regulates the activity of factors that enable the proper functioning and survival of β cells. It has been shown to negatively affect PDX1 (duodenal pancreatic homeobox-1), a key transcription factor important for β-cell development and function. The high activity of MST1 contributes to the phosphorylation of PDX1 (at the T11 site), leading to its degradation by the proteasome machinery, preventing it from acting as a transcription factor in the nucleus. PDX1 degradation occurs along the ubiquitin-proteasome pathway. Unfortunately, the final consequences of a deficiency in PDX1 function are manifested in impaired insulin secretion and the development of diabetes mellitus. Knockout MST1 in mice normalizes PDX1 under diabetic conditions [64], and showed, in these animals, normal blood glucose levels and elevated levels of circulating insulin in vivo and in vitro analyses. In addition, PDX1 target genes were normalized and GLUT2 localization is conserved. Additionally, the influence on the implementation of the phosphatidylinositol 3 kinase PI3K/AKT pathway, which is significantly related to the survival of β cells, was noticed. Serine-threonine kinases (AKT) regulate a variety of cellular events, including metabolism, through the phosphorylation of several other substrates. Their key functions indirectly influence the efficiency of functions and the survival of secretory cells. The split form of MST1, the active form, acts as an AKT inhibitor. By contrast, AKT-mediated phosphorylation negatively regulates MST1 activity and interferes with nuclear translocation [64,100].

#### Action of MST1 in Diabetes

The knowledge base accumulated in the basic research characterizing the properties of MST1 is a valuable starting point and the basis for designing therapeutic strategies depending on its inhibition. In recent years, in vitro and in vivo studies on various research materials, including pancreatic islets obtained from animals and patients with type 2 diabetes, have shown that pre-diabetes conditions and those that reflect full-blown diabetes activate MST1. The presence of chronic diabetic lesions strongly increases the autophosphorylation of MST1 (T183) and induces programmed cell death [44]. It is worth recalling here that the overexpression of MST1 induces apoptosis in a manner that is independent of diabetogenic conditions [64]. The above observations directly indicate that the main core kinase of the Hippo pathway is a versatile molecular tool and a valuable therapeutic tool [101]. The observed effects related to its action in diabetic states disrupt the secretion and action of insulin and, consequently, many systemic changes that result in β-cell apoptosis [64]. In fact, as a central mediator of the Hippo pathway, it influences various mechanisms of apoptosis. The therapeutic benefits and hopes apply not only to pancreatic β cells in diabetes, but may also apply to diabetes complications, including nephropathy and cardiomyopathy [43]. The loss of kinase (MST1-knockout (MST1-KO) mice) prevents the progression of diabetes and restores normal cell function [64]. 

A full description of the functions of the Hippo pathway will facilitate the selection of factors with which it will be possible to design and implement breakthrough therapies for various diseases, including diabetes. The starting point for the formulation of assumptions regarding the modulation of the Hippo pathway signals in diabetes is the use of factors with a protective effect on pancreatic β cells. This conclusion is based on the results of genetic MST1 deficiency, which brought beneficial effects in the form of glycemic normalization and an improvement in β-cell function, survival, and the inhibition of diabetes development [64]. The development of drugs targeting the MST1 kinase is ongoing, and so far no drugs have been approved to officially perform this function. The search for inhibitors that will function as specific drugs in the future generally involves two extremely different approaches. The first is the synthesis of new selective molecules dedicated to the pharmacotherapy of diabetes; the second is the search for alternative indications for previously used medicinal substances in the pharmacotherapy of other diseases. The advantage of the second strategy is the well-known safety profile of the drug, which consequently significantly reduces the time needed to develop a new drug form. Several small-molecule MST inhibitors have been described and their action profile has been established in the preclinical phase [76,102]. A common denominator of MST1 inhibitors (presented in this review) is their documented ability to reduce activity in the pancreas. The presented group of small molecules is not related pharmacologically, and their action has been presented at various stages of research advancement. This is why the form of their presentation retains an independent, untypical character for discussion.

Among the most advanced studies are those focusing on the action of neratinib, which is an inhibitor of tyrosine kinases and is used in anticancer therapy [44,101]. Multiple-target tyrosine kinase inhibitors (TKI), as the second group of small molecules (with a wide range of factors available), have been reported for years as potential candidates for the treatment of symptoms of diabetes mellitus [103,104]. Clinical trials have shown a beneficial anti-diabetic effect in patients with type 2 diabetes whose symptoms were alleviated by imatinib, sunitinib, desatinib and elrotinib [105,106,107,108,109]. Although the main goal is currently cancer therapy, and their action is based on ERBB2/EGFR inhibition, it has been noticed (in clinical observations) that they additionally alleviate the symptoms of diabetes [110,111]. Increased EGFR contributes to insulin resistance and correlates with a higher risk of developing diabetes, and the use of TKI inhibitors in most cases led to an arrest of diabetes development, normalization of glycemia, and in extreme cases even to diabetes re-emission. Some of the TKIs have already been thoroughly verified and approved by the FDA, which is highly beneficial [104,111]. Preclinical studies of these factors have demonstrated an inhibitory effect on pancreatic β-cell apoptosis in animals and humans, thus eliminating one of the main causes of diabetes [112]. The intensive verification of the effects associated with the presence of neratinib showed a strong inhibition of MST1, which resulted in an improvement in the survival of β cells in vitro and became the basis for the observation of molecular consequences [44]. By contrast, in vivo analyses have shown improvement in insulin resistance by inhibiting the production of tumor necrosis factor alpha (TNF-α) or by reducing endoplasmic reticulum stress [113,114,115]. Its main use is in blocking the growth of the human epidermal factor 2 receptor (ERBB2, HER2) and the epidermal growth factor receptor in the treatment of breast cancer [116,117,118]. It was approved as a drug by the Food and Drug Administration (25 February), in 2020, as NERLYNX, (Puma Biotechnology, Inc. CA, USA). The presented clinical trials of this drug (https://clinicaltrials.gov, state on 23 October 2021) include 69 studies at various stages of implementation and advancement (early phase 1, phase 3). Although clinical trials targeting cancer have shown a potential anti-diabetic effect, the standard analysis focusing solely on diabetes parameters has not yet started. It seems, however, that such research will appear in the near future.

Basic research focused on the use of neratinib as an MST1 inhibitor in the Hippo pathway under various experimental conditions, including diabetic conditions, shows promising prospects and relief of underlying symptoms. No toxic effects were observed during the performance of screening analysis to select the effective concentration for studies with isolated pancreatic islets (25 µM neratinib) and for intraperitoneal injections (marix-assisted laser desorption ionization imaging mass spectrometry (MALDI-IMS)). The specificity of the action on human and murine pancreatic β cells revealed an anti-apoptotic effect and increased survival, reflecting the action of neratinib in diabetogenic states. The analysis carried out did not reveal any side effects, including a reduction in glucose levels or β-cell function in the control animals. In addition, under high-glucose conditions, neratinib restored PDX1 expression and increased cell viability. Caspase-3 activity reflecting ER stress (experimentally induced) was also suppressed by neratinib in a dose-dependent manner, highlighting the many benefits of inhibiting pre- and diabetic-activated MST1. In addition, neratinib was effective in counteracting stress-induced MST1 activation, and apoptotic effectors cleaved caspase-3 and PARP. Similar observations, i.e., the inhibition of proinflammatory cytokine activation, MST1, and caspase-3 activation, were demonstrated in human islets incubated with neratinib under high-glucose/palmitate conditions. Comparable observations have been made in studies conducted in experimental MST1 knockout mouse models. Despite potent activating factors of apoptotic pathways (including pro-inflammatory cytokines and glycolipotoxicity), the pancreatic islets of MST1-KO mice did not show apoptosis, unlike WT. Under diabetogenic conditions, neratinib reduced apoptosis but exerted no significant effect in MST1-KO mice. The course of the apoptotic pathway under conditions of MST1 overexpression was successfully blocked by neratinib in in vitro experiments performed on the β cells of isolated human islets. Blocking the apoptotic pathway or alleviating the symptoms and causes of diabetes are groundbreaking and fundamental goals in diabetes management. Many cases have already been reported for neratinib. Nevertheless, it is important that the effects obtained at the laboratory level prove to be durable and fully recognized at the molecular level. Therefore, studies were also performed to identify the molecular identification triggered by neratinib. A real-time analysis of INS-1E cells with adenoviral overexpression of MST1/LATS2 transfected with LATS-BS revealed specific apoptotic consequences dependent on these two essential proteins in the Hippo pathway, disclosing MST1-LATS2’s interoperability in activating downstream processes. Only neratinib (compared to the lack of effect of the EGFR inhibitor canertinib) in this experimental system strongly inhibited both kinases, thus blocking the continuation of this process. In experiments conducted on mice with experimentally induced MLD-STZ type 1 and type 2 diabetes, neratinib improved glycemia, insulin secretion, and β cell survival in a mouse model of MLD-STZ diabetes (type 1). In models reflecting the state of type 2 diabetes, the islet architecture of the mildly dose MLD-STZ diabetic mice is characterized by structural abnormalities, including a reduction in both the number of β cells and the expression of important markers of glucose metabolism in cells that retain secretory function (PDX, NKX6.1, and the glucose transporter, GLUT2). Neratinib restored cell function and increased cell survival. Additionally, it restored the virtually lost expression of NKX6.1, as well as PDX1 and GLUT2. In another experiment, Ardestani et al. [44] showed that neratinib (5 mg/kg ip/31 days) in obese Lepr^db/db^ mice with diabetes counteracted high hyperglycemia. A stabilization of blood glucose levels was observed compared to the control group and no changes in body weight. In addition, glucose tolerance tests showed an amelioration of hyperglycemia and increased insulin secretion at all the measured time points in subjects receiving neratinib. In addition, it was shown that there was a slightly reduced ability to lower blood glucose in response to insulin. An analysis of β cell changes after neratinib therapy showed an increase in β-cell mass as an effect reflecting decreased apoptosis and increased proliferation (the markers Ki67 used and phospho-histone H3 (pHH3) immunolabeling). Moreover, an increase in PDX1 expression in cell nuclei was confirmed. Ex vivo experiments carried out on isolated pancreatic islets revealed a normalizing effect in severe states of diabetes. Their presence inhibited the activation of MST1 and the pro-apoptotic BIM protein and contributed to the full restoration of β cell survival. Additionally, an increase in PDX1 nuclear expression was observed. In another experiment, the effect of neratinib on the course of cytokine-induced β-cell apoptosis was analyzed. Pancreatic islets exposed to the cytokine mixture IL-1β/IFNγ showed a sharp increase in apoptosis; however, the restoration of β cell survival was observed in some experimental groups in the presence of neratinib [44].

The second factor that also shows promising prospects for action in diabetes is XMU-MP-1 [119,120]. Previous studies have demonstrated the pharmacokinetic efficacy of XMU-MP-1 in inhibiting MST1/2 in mice, further revealing liver and intestinal regeneration in vivo. The pharmacological manipulation of Hippo’s signaling pathway with XMU-MP-1 offers a broad spectrum and opens up new research opportunities in regenerative medicine, apoptosis, and certainly more, especially since the proposed inhibitor appears to be a safe targeted therapeutic agent in the treatment of tissue damage. The action of XMU-MP-1 mainly targets MST1/2 and inhibits both kinases with IC50 values of 71 nM and 38 nM, respectively, although the inhibition of MOB1 protein phosphorylation was observed in a dose-dependent manner [120]. The incubation of INS-1 cells with increasing doses of XMU-MP-1 (1–5 µM) showed inhibition of MST1/MST2. The reduced phosphorylation of other MST1/MST2 substrates, i.e., LATS1 and MOB1, was disclosed. Moreover, a dose-dependent increase in YAP activity was observed [119,120]. The MTT test showed that lower doses of XMU-MP-1 are not toxic to INS-1 cells, although its concentrations in the range of 3–5 µM decreased cell viability, and the highest dose of 5 µM caused the loss of ~30% of INS-1 cells (incubation 24 h). XMU-MP-1 at various concentrations of 1–3 µM increased the survival of INS-1 cells exposed to STZ (1–2 mM for 24 h). In in vivo studies, subjects with fully developed diabetes (STZ, 50 mg/kg body weight) were administered XMU-MP-1 (at a dose of 1 mg/kg/day) or DMSO (control group) for 21 consecutive days. Fasting blood glucose analysis as well as glucose tolerance test (GTT) or initial body weight monitoring showed no difference in blood glucose levels between the XMU-MP-1 group and the DMSO group. A reassessment (after 35 days) of fasting glucose and the performed glucose tolerance test showed positive effects of XMU in both groups of animals with experimentally induced diabetes (mild and severe form) compared to the control group (DMSO). The speed of regulation of blood glucose (15 min from intraperitoneal administration) to intraperitoneal glucose was also noteworthy. Blood glucose analysis in animals treated with XMU stabilized in a much shorter time than in the control group. The GTT results showed that the beneficial effects of XMU-MP-1 were only seen in the severe diabetic group. An analysis of the histological sections of the Langerhans pancreatic islets from the mice treated with XMU-MP-1 showed no statistically significant differences in the mean number of pancreatic islet cells or pancreatic islet area in nondiabetic and experimentally induced diabetic mice. A detailed analysis of the preparations from severely diabetic animals showed the beneficial effect of XMU-MP-1, which was manifested by an increase in the number of cells and islet area. In vivo studies of the effectiveness of the XMU-MP-1 inhibitor showed its greater effectiveness only in the severe diabetes group [119]. 

Another MST1 inhibitor is Asialo-erythropoietin [121], an EPO derivative with no sialic acid residue. It does not exhibit hematopoietic properties and is characterized by a very short circulating half-life [122]. The proposed molecule is a cytokine with a broad neuroprotective effect [123]. Among the group of derivatives of its group, it shows the best protective effect against the diabetes-induced apoptosis in Rin-m5F cells; therefore, its action is presented in this review. In in vitro experiments, RIN-m5F cultured cells were incubated with both asialo-rhuEPO^P^ at various concentrations (20, 40, 60, 80, or 100 IU/ml) and STS (staurosporine) for 24 h. The changes in cytotoxicity in the control group cellstreated with STS alone amounted to 46%, while the simultaneous exposure of STS and 20–100 IU asialo-rhuEPO^P^ caused reduced cytotoxicity (in the range of 40–20%). The cytoprotection index revealed the highest values for the doses of 60, 80, or 100 IU/mL asialo-rhuEPO^P^ and was 41–56%. The protective effect of asialo-rhuEPO^P^ is due to its ability to inhibit MST1 and caspase-3 under diabetic conditions, which was confirmed by a significant reduction in the amount of cleaved MST1 and cleaved caspase-3 compared to control cells (incubation only with STS). Asialo-rhuEPO^P^ under the conditions of induced diabetes reduces the expression of the pro-apoptotic protein Bax and increases the amount of the anti-apoptotic Bcl-2, which was demonstrated in the analysis of the expression profile of mitochondrial proteins. Additionally, its anti-apoptotic effect is related to the preservation of AKT phosphorylation. Moreover, it increased the amount of PDX-1 protecting it from degradation, and the obtained value is comparable to the control levels. By contrast, the decreased insulin secretion (30%) due to STS was restored by asialo-rhuEPO^P^ to approximately 22% in Rin-m5F cells [121].

### 4.2. Large Tumor Supresor1/2 (LATS 1/2)

The large tumor suppressor (LATS) gene family, including LATS1 and LATS2, an MST1 downstream substrate, is an essential component of the Hippo pathway [124]. These proteins are critical regulators of cell apoptosis, originally identified in fruit flies [125]. It belongs to the serine-threonine protein kinases. Human LATS1 and LATS2 belong to the AGC subfamily, related to Dbf2-related nuclear kinases (NDR1/2) [126]. Both kinases show significant homology (85% similarity) in the sequence of the kinase domain located at the C-terminus of the proteins. By contrast, the N-terminal is much less conservative [127]. The structure includes: C-terminal kinase domain, protein binding domain (PBD), two conserved LATS domains (LCD1 and LCD2), ubiquitin binding domain (UBA) and at least one PPxY motif (P: proline, X: any amino acid, Y: tyrosine), which binds to the YAP and TAZ domains. The C domain features two conserved Ser/Thr phosphorylation sites. The site of autophosphorylation is serine, while threonine is the site of phosphorylation by MST1 and MST2. At the amino terminus (N), there are two stretches of conserved sequences (LCD1 and 2), and their presence is essential for the proper regulation and function of the core kinases of the Hippo pathway [128]. Moreover, both LATS1/2 within the N terminus exhibit evolutionarily conserved ubiquitin-related domains. The presence of such domains enables the binding of the ubiquitinated protein and may therefore be involved in their activation [126,129]. Both isoforms of kinases possess unique features in their structure, which are the basis of various protein–protein interactions. At its N-terminus, LATS1 features a proline-rich domain, while LATS2 contains the unique PAPA repeat. Additionally, both kinases were observed to code for the PPxY motif that determines interaction with downstream components of the Hippo pathway, including YAP and TAZ [130]. The hydrophobic motif of both kinases is similar to other AGC kinases such as AKT, S6K1 (important fo mTORC1) and PKC [52]. LATS1 and LATS2 constitute a unique family of signaling in the cell; they are important signaling molecules, and their action is aimed at the regulation of transcription, the maintenance of genetic stability, the modulation of cell cycle checkpoints, and the induction of apoptosis [131,132,133]. Moreover, these proteins are involved in the regulation of cell proliferation and apoptosis; therefore, a conservative belief prevails, which is not entirely correct, that they are mainly involved in the development of cancer [131]. The tremendous metabolic plasticity of pancreatic β cells and their stress response followed by pro-diabetic signals are driven by the structure and dynamics of complex signal transduction networks. These factors influence the components of the Hippo pathway by modulating it. Studies have shown that Nf2 is an upstream regulator of the Hippo pathway’s signaling [134]. It was found to be expressed in cellular models-INS-1E, as well as in primary human islets. The loss of Nf2 function in pancreatic cells can overcome cell apoptosis and is revealed by the inhibition of LATS2 activity. This molecular modulation exerts no effect on cell function as well as on the functional genes of the cell identity [73]. LATS-regulated cell apoptosis also affects downstream targets, the most common of which are P53, FOXO1, abelson tyrosine kinase c-Abl (c-Abl), and YAP [135,136,137]. LATS is a critical component of the Hippo pathway for the induction of apoptosis in β cells [46]. Its mechanism of action is based on the mechanistic target of rapamycin 1 (mTORC1). The central kinase responsible for regulating cell growth in mammals is the mechanistic target of rapamycin, also called the mammalian target of rapamycin (mTOR). So far, two different complexes have been described: mTORC1 and mTORC2. The difference between them is due to the presence of unique helper proteins, Raptor and Rictor, respectively [138]. mTORC1 is responsible mainly for the regulation of cell growth and metabolism; mTORC2 is responsible for the control of proliferation, as well as cell survival [139]. The mTOR pathway is activated by energy input, especially in two important forms: amino acids and insulin-like growth factors (IGFs). It is thought to mediate β cell growth and expansion [138]. However, its excessive activation and negative effects related to the loss of β cells were revealed in animal models showing the presence of type 2 diabetes [46], clearly pointing to an ongoing apoptosis scenario, in which the presence of conditions leading to diabetes activates the sequelae cascade, i.e., diabetic conditions activate the LATS and thereby activate mTORC1 by suppressing AMP-activated protein kinase (AMPK) signaling. Moreover, growing evidence suggests that AMP-activated protein kinase (AMPK) and mTOR play a pivotal role in autophagy and apoptosis [140].

#### Action of LATS in Diabetes

Research focused on the role of LATS1/2 in the process of the apotosis of pancreatic β cells is still carried using a conventional understanding of the mechanism of regulation of this kinase in this process. Yuan et al. [46] showed that large tumor suppressor 2 (LATS2) induces apoptosis, but also affects pancreatic β cells by participating in the development of apoptosis. The action of this isoform is revealed in diabetic states. INS-1E cells and cells derived from human pancreatic islets have been shown to be highly susceptible to LATS2 deficiency, resulting in improved viability, increased cell mass, and restoration of insulin secretion, which combine to reduce the development of diabetes.

LATS1/2 kinase activity results in the phosphorylation of YAP targets on S127 (LATS specific site) resulting in the exposed docking site of the 14-3-3 proteins. As a result, YAP cytoplasmic sequestration occurs. LATS kinase inhibitors used so far have found application in research on neoplastic changes and proliferation. 

Few inhibitors targeting the LATS kinase have been designed so far, which is directly related to the lack of a fully known crystal structure of this kinase [141] and constitutes a serious obstacle to the formulation of a specific action molecule. More recently, information has appeared in research about a TRULI inhibitor obtained from Enamine LLC (Z730688380, Monmouth Junction, NJ, USA) and targeting LATS1 and LATS2 [141]. Relatively recently (WO/2018/198077), 6-6 fused bicyclic heteroaryl compounds have been patented, the recommended therapeutic applications of which are related to the healing of various wounds [142,143].

There are many articles in digital databases describing the function of LATS1/2 in pancreatic proliferation or cancer, which unfortunately does not apply to diabetes. To our knowledge, which is supported by a detailed analysis of the many available databases containing articles in the field of medicine and life sciences, so far, only one pioneering publication [46] detailing the operation of LATS has appeared. Extensive studies have been performed on β cells, isolated human and murine pancreatic islets, and diabetic mice or other genetically generated models. The interpretation of LATS2 action in diabetes and apoptosis has been confronted with an explanation of the role and interaction of mTOR and autophages in these processes. Despite the fact that there are many documented experiences regarding the function of the mTOR pathway itself and autophages in this specific area, it is difficult to refer to them as they do not directly discuss the function of the Hippo pathway, although it may be assumed that the effects obtained are due to its involvement.

LATS2, as one of the core kinases of the Hippo pathway, is responsible for interfering with the survival and functioning of β cells. This finding is confirmed by both in vitro and in vivo experiments, which have shown that diabetogenic states increase the activation of LATS. The effects of glucotoxicity and glycolipotoxicity were the direct cause of the induction of apoptosis and were also associated with loss of β cell function due to LATS1/2 overactivity. The increase in kinase activity produced a “ripple effect”, which was LATS-dependent: increased YAP phosphorylation and increased MOB1 levels. Other experiments on INS-1E and human pancreatic islets revealed an increase in MOB1 levels as a result of LATS2 overexpression alone (induced by adenoviral transduction). The overexpression of LATS causes apoptosis, as evidenced by the split caspase-3 and poly (ADP-ribose) polymerases (PARP), another caspase-3 substrate, present in the cell. Furthermore, in the isolated pancreatic islets of obese Lepr^db/db^ (db/db) mice, it was confirmed that the level of exogenously expressed pYAP and MOB1 proteins was increased. This finding was confirmed by experiments in which the loss of endogenous LATS2 activity protected the tested INS-1E cells against apoptosis, despite the diabetogenic conditions of their culture. This conclusion was supported by the results of an experiment in which the loss of endogenous LATS2 activity shielded and moderated the development of apoptosis in INS-1E cells, even indicating a pro-survival effect, despite the unfavorable diabetogenic conditions of their culture. In addition, β cells have shown a potential difference in the regulation of apoptosis between LATS1 and LATS2 [46].

Multiple pathways can be crucial for the induction and execution of the very complex process of apoptosis. Collective action is often filled with dependencies in the course of molecular mechanisms, creating the basis for mutual control. Research in the area of β cells is ongoing, and it is impossible to mention all the possible interactions; below, therefore, those whose operation is critical to revealing or modifying the LATS function are discussed. An example of such an interaction in the β pancreas is the mTORC1–Hippo LATS crosstalk. Yuan et al. [46] revealed that LATS2 acts as a key upstream activator for the mTORC1 pathway in β cells exposed to stress and that the knockout of Hippo kinase was significant for weakening downstream signaling, including that of mTORC1. The arrest of the mTORC1 pathway was revealed by the blocking of β cell apoptosis, which clearly showed that the pro-apoptotic effect of LATS2 is mTORC1-dependent and is over-activated by LATS2 in diabetes in INS-1E cells and human pancreatic islets [13]. The development of apoptosis in β cells and isolated human pancreatic islets grown under pro-apoptotic (glucotoxicity) conditions revealed the implementation of mTORC1 in a LATS2-dependent manner. The silencing of mTORC1 resulted in an arrest of apoptosis despite toxic pancreatic β cell toxic and LATS activation. On the other hand, it was revealed that the overexpression of LATS2 in these cells activated mTORC1. The action of LATS in diabetic conditions leads to apoptosis mediated by the activation of Rag-mTORC1 [46].

Autophagy is responsible for the functionality of cell organelles essential for the survival/functioning of β cells. Their function is also important in improving insulin sensitivity during a high-fat diet in mice; [144] in diabetes, however, it plays a role in its pathogenesis [145]. It has been shown that autophagy can protect diabetes in a manner independent of mTOR. On the other hand, both insulin and mTOR are inhibitors of autophagy. The haploinsufficiency of autophagy in murine animal models of obesity increased insulin resistance, which was accompanied by elevated lipid levels, as well as inflammation [145]. It has been observed that the elimination of both misfolded proteins and mitochondria to protect β cells from loss of function in diabetes is largely done by autophages [146,147]. Thus, loss of autophagy is what causes obesity-induced diabetes to develop. Ardestani et al. [148] demonstrate a strong functional relationship between mTOR and autophagy. The mechanism induced by insufficiently elevated levels of nutrients leading to long-term activation of mTORC1 likely leads to β cell failure, which is the opposite of how this pathway works normally. The prolonged state of increased mTORC1 activity is manifested by a progressive loss of function, β cell mass and associated mTORC2 impairment, and mTOR-mediated autophages [148]. 

Studies focused on the role of LATS2 as an activator of mTORC1 revealed an impairment of autophagy in β cells, which led to the reduced viability of these cells and, at the same time, revealed another potential mechanism of counteracting. In vitro experiments revealed apoptosis in INS-1E cells and isolated human islets by simultaneous inhibiting autophagy and overexpression of LATS2. Under the conditions of LATS2 silence and autophagy inhibition, the course of apoptosis was reduced. As apoptosis increased, LATS2 decreased autophagic flow in β cells and human pancreatic islets. The accumulation of autophagic flux markers (1A/1B-light chain 3 (LC3-II) and p62 (SQSTM1)) under LATS2 overexpression is also evidence of attenuated autophagic flux dependent on increased Hippo core kinase activity. Moreover, p62 is an integral part of the mTORC1 complex and it interacts in an amino acid-dependent manner with mTOR and the raptor, confirming the mediation of this pathway between LATS2 and autophagy. Conversely, the loss of LATS2 in human islets decreased the accumulation of LC3-BII and p62 proteins caused by autophagy inhibitors. The above data prove the significant role of LATS2 in autophagy flow and, at the same time, indicate a participation in the implementation of apoptosis induced by defective autophagy. It is also interesting that LATS2 is located in lysosomes and acts as a substrate for autophagy in the degradation process. Moreover, it was confirmed that in the pancreas, under the conditions of increased LATS2 in the pancreatic β cells, macroautophagy dominates as one of the two mechanisms of autophagy [46]. 

In vivo experiments were carried out on a mouse model with a β cell-specific LATS2 deletion (β-LATS2-/-) and the LATS2fl/fl control group with MLD-STZ experimentally induced diabetes. While control animals showed a progressive development of diabetes (hyperglycemia, severely impaired glucose tolerance, impaired insulin secretion), β cell LATS2-deficient mice showed milder changes in diabetes development induced by low doses of STZ.

An analysis of β cell mass quantification on histological slides in β-LATS2-/- and LATS2fl/fl mice showed a significantly reduced β cell mass in control mice. Moreover, further analysis showed that the pancreatic β cell mass of LATS2-ablated mice were a result of increased proliferation and reduced apoptosis. The above results clearly indicate that the deficiency of LATS in the pancreas exerts a protective effect on β cells in diabetes [46]. Similar results were obtained in animals fed on a high fat/high sucrose diet for 17 weeks, which in control animals contributed to the development of insulin resistance, β cell failure, and hyperglycemia. The development of symptoms was accompanied by diet-induced LATS2 activity. Animals with the LATS-/- genotype under the same experimental conditions showed significantly improved glucose tolerance, higher insulin secretion, and a positive proliferation/apoptosis ratio compared to control animals. Moreover, immunohistochemical analyzes of the β islet cells of high-fat/high-sucrose-fed mice showed a significantly higher expression of the phosphorylated-form ribosomal protein S6 (pS6) which is a reliable marker of mTORC1 and S6K activation compared to the group of control animals. Different results were observed in HFD-fed β-LATS2-/- mice, in which pS6 expression was normalized. LATS2 deficiency blocks mTORC1 implementation in high-fat/sucrose fed mice and maintains the interaction between LATS2 and the mTORC1 autophagy axis [46].

All the above studies provide the basis for the conclusion that both the Hippo-LATS and mTORC1 pathway kinase, as well as autophagy, co-create signaling related to pancreatic b cell survival susceptible to stress factors. The protective effect of autophagy is based on the positive feedback directed to LATS2 and the cell cover under conditions of severe stress. Nevertheless, such a mechanism of action does not function in long-term stress (e.g., diabetes), the effect of which is the development of apoptosis through the activation of LATS2, mTORC1, which results in the implementation of defective autophagy. In conclusion, LATS2 may be a second therapeutic target alongside MST1, whose blockage of action may lead to improved function and increase β cell survival in diabetes [46].

### 4.3. Yes-Associated Protein (YAP) 

Two YAP isoforms (YAP1 and YAP2) have been described so far [149]. They show some structural differences related to the WW domain, the presence of one described for YAP1 and two for YAP2. In both the N and C terminal isoforms a proline-rich region, TEAD binding domain, SH3 binding motif, transcription activation domain, PDZ binding motif have been recognized [150]. In addition, YAP as a transcriptional co-activator features a transcription activation domain but no DNA binding domain [151]. The YAP protein is considered to be an analog of TAZ; their functions are referred to as redundant and are usually formulated as YAP/TAZ [152,153]. Although they may function independently, YAP is considered to exert a stronger effect than TAZ. Similarities in topological structure and the presence of about half of the identical oxygen sequences were recognized between YAP1 and TAZ [153]. The major regulator of YAP/TAZ through phosphorylation is the kinase cascade of the Hippo pathway [154]. The phosphorylation of YAP and TAZ (= inactivation) by LATS1/2 (Yap on S127 or TAZ on S89) results in the binding of the 14-3-3 protein and the sequestration of YAP in the cytoplasm results in ubiquitin-dependent degradation [82]. When the Hippo pathway kinases are inactive, the downstream effectors, YAP and TAZ, are located in the nucleus and activate transcription programs in cooperation with the transcription factors of the TEA domain (canonical pathway) [155]. Research data indicate that there are more transcription factors in addition to the TEAD family mentioned above, including P73 and Runx2 cells, causing proliferation and survival [84]. The participation of AMPK was also found, the effect of which in this area is inhibition of cell growth, and due to YAP phosphorylation it leads to the disruption of YAP-TEAD [156]. The scope of its action is very wide and is mainly related to driving embryonic growth [157,158], stem cell proliferation and differentiation [159,160], and vascular remodeling [161], and it plays an important role in the development of diseases of the nervous system and cancer [162]. Moreover YAP/TAZ stand out as elements of the Hippo cascade involved in the regulation of organ location and size, as well as tissue regeneration [163,164]. Thus, YAP/TAZ dysfunction is causing an increasing number of human diseases [165]. Although its basic action is still understood stereotypically and associated mainly with embryogenesis or cancer, in the last few years there were proposals to use its potential in the treatment of patients with diabetes, hypertension, and metabolic syndrome in the prevention of heart failure syndromes [166]. For example, manipulating the activity of YAP, which is considered by many to be an oncoprotein, is not safe without taking appropriate precautions, including targeted action or activity control. The YAP function changes drastically depending on the type of stress [166], and also shows significant sensitivity to high/low glucose levels in insuli target cells [167]; therefore, its mechanism of action should be carefully investigated. It was also observed that it was activated in patients with heart failure and diabetes [166]. Other studies have shown that YAP plays an important role in both the replication and survival of β cells. YAP-induced β-cell proliferation has been diagnosed in adult studies and involved in the protective functions against the pro-apoptotic effects activated by diabetes [168]. All indications show that YAP activity is perceived as timely and activated under conditions of demand, e.g., the regeneration of β cells damaged in diabetic conditions. The relationship with many physiological processes, including growth stimulation, the regulation of the cell cycle, or proliferation, is a kind of functional showcase of YAP and evidence of interaction with many signaling networks [168,169,170,171,172,173]. The forkhead box M1 protein (FOXM1) was also noted in the YAP-dependent process of pancreatic β cells proliferation. FOXM1, as a member of the FOX family, plays an important role in the progression of the cell cycle, and its significant effect has also been noticed in postnatal proliferation and mass expansion of β cells exposed to increased stress in vivo [168,174]. Oxidative stress, one of the many factors leading to the deterioration of β cell function during the development of type 2 diabetes, can be prevented by the action of thioredoxin (Trx), a redox protein expressed in pancreatic β cells. Most likely, the operation of Trx1 is YAP-dependent, and its participation in specific regulatory signal loops is complex. In vivo studies in rodent models with type 1 and type 2 diabetes have shown that β-cell-specific Trx1 overexpression inhibits their death and slows the progression for type 1 diabetes [175,176]. By contrast, silencing Trx1/2 lowered exogenously expressed YAP. On the other hand, the induction of YAP overexpression in healthy and diabetic human islets and cultured β INS-1E cells resulted in an increase in Trx1/2 activity.

## 5. Conclusions

According to the data above, a positive answer can be given to the question in the title, “Apoptosis in type 2 diabetes: Can it be prevented?”. Using the Hippo pathway as a tool, the new strategy for treating diabetes should guarantee the viability and functionality of the cell and organ and not disturb the homeostasis of the organism. Current treatment for type 2 diabetes does not inhibit apoptosis, does not protect β cell mass and is associated with a loss of secretory function. The results of basic research on the effects obtained from modulation of Hippo pathway signaling allow the design of a breakthrough therapy that addresses all the shortcomings of the current treatment regimen. Studies of human β cells/pancreatic islets under experimentally induced diabetes have shown that the inhibitory effects of MST1 and LATS2, the major kinases of the Hippo pathway, exhibit high functional and anti-apoptotic efficacy [43]. This partially explains why symptoms of type 2 diabetes can be alleviated in some cancer patients taking tyrosine kinase inhibitors (for MST1). Therefore, research in this direction intensified; as a result, the first small molecules were selected. The effect of the strong MST1 inhibitor neratinib (the most widely controlled), documented in basic studies, is the basis for further research aimed at implementing a new diabetes treatment formula based on the maintenance of β-cell survival and function and the elimination of hyperglycemia [44,101]. Perhaps in the near future, further basic research may reveal how β-cell metabolism changes with long-term treatment with neratinib. And although this drug has been approved for oncological therapy, the spectrum of its long-term consequences has not yet been demonstrated. Therefore, this should be carefully considered, especially with regard to side effects at the cellular and systemic levels. Is the approved route of taking the drug the most effective way to control the effects of diabetes? How long will the path’s sensitivity to its operation last? Furthermore, although research into the pathological state of diabetes (at the level of basic research) has produced many desirable effects, special attention should be paid to the physiological aspects that may modulate the functions of this pathway. There are still many open questions, especially with regard to the factors modulating the signals up and down the Hippo path. Unfortunately, it should also be remembered that theoretically, the inhibition of MST1 will not stop the Hippo pathway, and cell stress factors (those that lead to diabetes) may activate the second kinase of the LATS2 pathway (e.g., via Merlin) [46,65]. Perhaps future research will need to consider bidirectional inhibition also targeting LATS2 kinase. What will be the consequences for the β cell? Research into a selective LATS2 inhibitor is still ongoing. It is only a matter of time before these questions are answered.

Promising preliminary results from in vitro studies revealed that the high level of YAP expression in pancreatic islets maintains insulin secretion and normal gene expression. Nevertheless, the target genes for YAP-TEAD emerge in the form of a very diverse function, as revealed, in this context, by the Hippo pathway, particularly proliferation, control of homeostasis at the tissue level, miRNA biogenesis, control of organ size, etc [168]. The mechanism of action is extremely sensitive and requires the absolute recognition of the interactions upon which its function may depend. On the other hand, the manipulation of YAP activation may turn out to be specific for the restoration of the pool of lost cells in diabetes. Therefore, basic analyses in this area will certainly be extended to in vivo experiments. Therefore, we must continue to wait for the full picture of the effects to emerge.

As demonstrated above, the Hippo pathway offers significant potential in the future treatment of type 2 diabetes. Using its possibilities will certainly break the usual patterns of diabetes treatment. Modulations in its activity at different levels of core kinases may inhibit apoptosis and, thus, the loss of insulin-secreting cells, while restoring their full functionality.

## Figures and Tables

**Figure 1 ijms-23-00636-f001:**
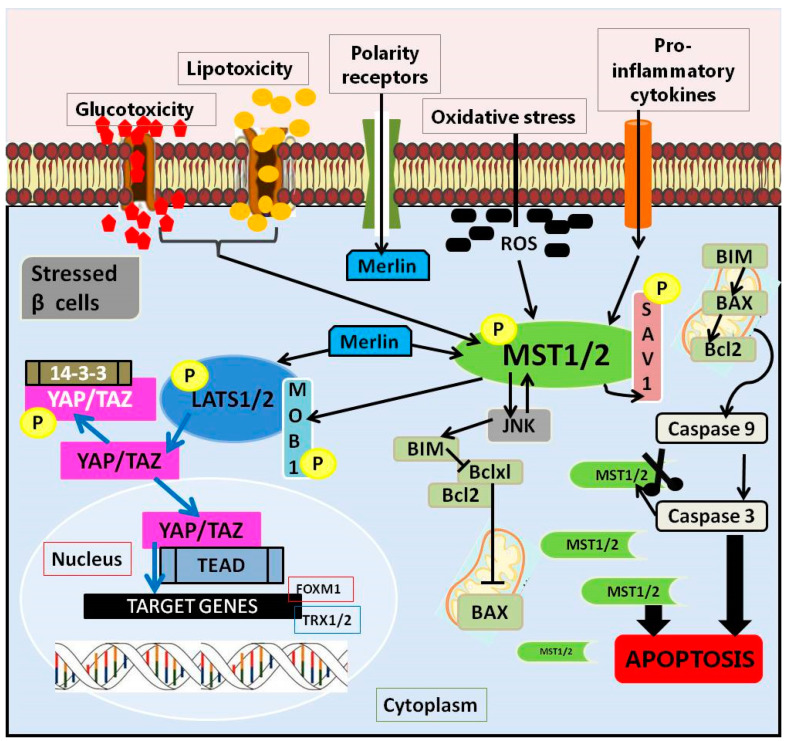
The Hippo pathway in pancreatic β cells.

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
