# Peer review of "Apoptosis in Type 2 Diabetes: Can It Be Prevented? Hippo Pathway Prospects"

_ijms, 2022, doi:10.3390/ijms23020636_

Round 1

Reviewer 1 Report

This review, by Kilanowska et al., attempted to summarize the regulatory functions exerted by the HIPPO signaling pathway in the development/treatment of type 2 diabetes through its regulation of pancreatic b cell apoptosis.

My major critique is:  

The manuscript piled a lot of different writing styles from the references cited by the authors, often redundant, un-organized, and inconsistent.  There are no insights or useful suggestions/discussions other than simply piles of publications.

My minor critique is:

There are multiple obvious editorial mistakes, for instance: “type2 diabetes” in places were written reversely as “diabetes type 2”; gene NF2 product Merlin, was interchanged freely even in the Figure1, which makes reading difficult; the mention of “HIPPO CORE kinases” but not defined what they are when it first appeared; similarly, the definition of “LATS1/2” appeared only late on p12 …… so on and so forth. 

Overall, I think the review is of poor quality and does not provide a clear current knowledge of the topic, and does not include any thoughtful insights and/or possible clinical directions.

Author Response

Dear Mr./Mrs.

Thank you for your comments.

I kindly inform you that the suggested corrections have been incorporated into this text. The correctness of the citations cited in the text was verified, and in cases requiring it, corrections were made. The manuscript introduction was simplified and shortened. In addition, the nomenclature was standardized.

In response to the objection to the lack of conclusions in the manuscript, I clarify that the study raises conclusions for all of the Hippo core proteins. Each time they constitute a summary of the presented research described in source publications:

Conclusions

Number of page

Number of verse

References

Hippo

2

32

Ardestani et al. 2019; Ardestani et al. 2016; Yuan et al. 2021

3

5

Avruch et al. 2012

Structure and action

3

7

Saucedo et al. 2007; Zhao et al. 2010; Matallanas t al. 2008; Ardestani et al. 2016; Dong et al. 2009; Ardestani et al 2018; 2015; Fallahi et al. 2016; Maugeri-Saccàa et al. 2018; Holden et al. 2018; Misra et al. 2018; Zanconato et al. 2018; Wu et al. 2021; Iwasa et al. 2018;

Merlin

5

12

Yuan et al. 2016; Yuan et al. 2017

MST1

7

7

Ardestani et al 2018; Ardestani et al 2016; Ardestani et al 2014; Rahmani et al. 2009; Lei et al. 2003;

MST1 inhibitors

7

33

Lee et al. 2001; Graves et al.  1998; Ardestani et al. 2018; Ura et al. 2001;

Ardestani et al. 2019a; Ardestani et al. 2019b; Faizah et al. 2020; Arthur et al. 2017;

LATS

12

15

21

26

Ke et al. 2004; Aylon et. al. 2010; Shao et al. 2014; Bu et al. 2020; Yuan et al. 2021

Yuan et al. 2021

YAP

16

26

Ikeda et al. 2019; Sayedyahossein 2020; Yuan et al. 2016; Zhou et al. 2021; Moroishi et al. 2015; Panciera et al. 2017; Piccolo et al. 2014; Mugahid et al. 2020;

The manuscript is a review of analyzes already carried out in this area, for which the author expressed his approval and repeatedly emphasized their therapeutic potential.Therefore, my own conclusions were limited to the above.Perhaps the previous form of the manuscript, which the reviewer found chaotic, made the presented conclusions invisible.

On the objection:

"(...) I think the review is of poor quality and does not provide a clear current knowledge of the topic, and does not include any thoughtful insights and/or possible clinical directions (...)"

all available studies were analyzed. They show that modulation of the Hippo pathway in the region of MST1 and LATS1/2 inhibition leads to restoration of β cell function and apoptosis arrest. However, research at the clinical level has yet to be done. The conducted review of the available publications leads to the conclusion that there is a lack of full clinical trials in this area. Therefore, when asked, "Apoptosis in type 2 diabetes, can it be prevented?" should be answered positively. This thesis is supported by the cited human islet/cell studies and studies mainly on neratinib and other tyrosine kinase inhibitors (as described side effects in cancer treatment), but clinical studies are required to fully support this hypothesis. The above conclusion is reflected in the manuscript. On the other hand, in connection with the allegation that there is no clear current knowledge, I would like to point out that over 50% of the cited publications were written in the last 5 years, and the conclusions contained in them are the basis for the latest research.

Yours sincerely.

Agnieszka Kilanowska

Reviewer 2 Report

This is a well written review article about the hippo pathway as a target to treat type 2 diabetes.

There are some typographical errors that need to be corrected.

The authors might include a small chapter about the link between hippo pathway alterations and gut microbiota in T2DM (e.g. PMID:33223509).

Author Response

 Dear Mr./Mrs.

Thank you for your comments.

As suggested, I have included a short fragment related to the recommended research, which will certainly allow to popularize the theses contained in it (page 2, verse 35-37).

"Primary studies have shown that abnormal expression of the Hippo signaling pathway in tested peripheral tissues may initiate type 2 diabetes and be associated with the gut microbiota as well as with the aging process."

Yours sincerely.

Agnieszka Kilanowska

Round 2

Reviewer 1 Report

I see the authors have made some good revisions regarding structure and some of the grammatical changes of this manuscript.  There are still some places that needs to be fixed, see below:

  • Intro, line 2, “popular” should be “prevalent”. A “popular” disease is not a fitting way of expression.
  • Section 2 title: should be “How exactly it works?”
  • Section 3, define “Hoppo core kinases” first before discussing their functions
  • Fig1, most of names used in the figure are protein names (Bcl2, Caspase3,9, LATS etc.).   NF2 is the gene name of Merlin.  When gene name is different than the protein name, use the protein name (Merlin) in the figure for function description
  • 1.1.2, it has been “disclosed” should be changed to “shown”, or “revealed”

The authors replied to my point of “thoughtful insights and clinical directions” in a way by saying that they have done a thorough citation check and included the most up-to-date publications.  These are two different things, however.  What the authors should add/discuss after piling up the published references are their own thinking and insight based on this available knowledge.  A good review article should be up to date with knowledge, while also thought provoking.  

Author Response

Dear Madam/Sir,

Thank you very much for your valuable comments.

In the attachment, I am sending the paper with the introduced corrections (also applies to the diagram).

I would like to inform you that in Chapter 2 I included a definition of the term "core of the Hippo pathway". (page, verse 11)

"The essential elements of the Hippo pathway referred to as the core of the Hippo pathway, are the mammalian sterile 20-like protein kinase 1 and 2 (MST1/2), kinase cascade and large tumor suppressors 1 and 2 (LATS1/2)"

As recommended, I expanded the Conclusions subsection. To facilitate access to the content of the text, I present it below.

Answering the question in the title: "Apoptosis in type 2 diabetes, can it be prevented?" - according to the above data, give a positive answer. Using the Hippo path as a tool. The new strategy for treating diabetes should guarantee the viability and functionality of the cell and organ and not disturb the homeostasis of the organism. Current treatment for type 2 diabetes does not inhibit apoptosis and does not protect β cell mass and is associated with a loss of secretory function. The results of basic research on the effects obtained from modulation of Hippo pathway signaling allow the design of a breakthrough therapy that addresses all the shortcomings of the current treatment regimen. Studies of human β cells/pancreatic islets under experimentally induced diabetes have shown that the inhibitory effects of MST1 and LATS2, the major kinases of the Hippo pathway, exhibit high functional and anti-apoptotic efficacy [43]. This partially explains why symptoms of type 2 diabetes can be alleviated in some cancer patients taking tyrosine kinase inhibitors (for MST1). Therefore, research in this direction was intensified, as a result of which the first small molecules were selected. The effect of the strong MST1 inhibitor neratinib (the most widely controlled one), documented in basic studies, is the basis for further research aimed at implementing a new diabetes treatment formula based on the maintenance of β-cell survival and function and the elimination of hyperglycemia [44,101]. Perhaps in the near future, further basic research may reveal how β-cell metabolism changes with long-term treatment with neratinib. And although this drug has been approved for oncological therapy, the spectrum of long-term consequences has not yet been demonstrated. Therefore, this should be carefully considered, especially with regard to side effects at the cellular and systemic levels. Is the approved route of taking the drug the most effective way to control the effects of diabetes? And how long will the path's sensitivity to its operation last? And although research into the pathological state of diabetes (at the level of basic research) has produced many desired effects, special attention should be paid to the physiological aspects that may modulate the functions of this pathway. There are still many open questions, especially with regard to the factors modulating the signal up and down the Hippo path. Unfortunately, it should also be remembered that theoretically, inhibition of MST1 will not stop the Hippo pathway, and cell stress factors (those that lead to diabetes) may activate the second kinase of the LATS2 pathway (e.g. via Merlin) [46,65]. Perhaps the research will need to consider bidirectional inhibition also targeting LATS2 kinase. What will be the consequences for the β cell? Research into a selective LATS2 inhibitor is still ongoing. It is only a matter of time before these questions are answered.

Promising preliminary results from in vitro studies revealed that the high level of YAP expression in pancreatic islets maintains insulin secretion and normal gene expression. Nevertheless, the target genes for YAP-TEAD emerge in the form of a very diverse function as revealed, in this context, by the Hippo pathway. In particular, proliferation, control of homeostasis at the tissue level, miRNA biogenesis, control of organ size, etc [177]. The mechanism of action is extremely sensitive and requires the absolute recognition of interactions upon which its function may depend. On the other hand, manipulation of YAP activation may turn out to be specific for the restoration of the pool of lost cells in diabetes. Therefore, basic analyzes in this area will certainly be extended to in vivo experiments. Therefore, we still have to wait for the full picture of the effects.

As demonstrated above, the Hippo pathway has a huge potential in the future in the treatment of type 2 diabetes. Using its possibilities will certainly break the usual patterns of diabetes treatment. Modulations in its activity at different levels of core kinases may: inhibit apoptosis, and thus the loss of insulin-secreting cells while restoring their full functionality.

I hope that the changes I have introduced will meet your expectations. Yours sincerely. Agnieszka Kilanowska 
